# Genome-Based Multi-Antigenic Epitopes Vaccine Construct Designing against *Staphylococcus hominis* Using Reverse Vaccinology and Biophysical Approaches

**DOI:** 10.3390/vaccines10101729

**Published:** 2022-10-16

**Authors:** Mahreen Nawaz, Asad Ullah, Alhanouf I. Al-Harbi, Mahboob Ul Haq, Alaa R. Hameed, Sajjad Ahmad, Aamir Aziz, Khadija Raziq, Saifullah Khan, Muhammad Irfan, Riaz Muhammad

**Affiliations:** 1Department of Health and Biological Sciences, Abasyn University, Peshawar 25000, Pakistan; 2Department of Medical Laboratory, College of Applied Medical Sciences, Taibah University, Yanbu 46411, Saudi Arabia; 3Department of Pharmacy, Abasyn University, Peshawar 25000, Pakistan; 4Department of Medical Laboratory Techniques, School of Life Sciences, Dijlah University College, Baghdad 10011, Iraq; 5Institute of Biological Sciences, Sarhad University of Science and Information Technology, Peshawar 25000, Pakistan; 6Institute of Biotechnology and Microbiology, Bacha Khan University, Charsadda 24840, Pakistan; 7Department of Oral Biology, College of Dentistry, University of Florida, Gainesville, FL 32611, USA

**Keywords:** *Staphylococcus hominis*, multi-epitopes-based vaccine, reverse vaccinology

## Abstract

*Staphylococcus hominis* is a Gram-positive bacterium from the *staphylococcus* genus; it is also a member of coagulase-negative *staphylococci* because of its opportunistic nature and ability to cause life-threatening bloodstream infections in immunocompromised patients. Gram-positive and opportunistic bacteria have become a major concern for the medical community. It has also drawn the attention of scientists due to the evaluation of immune evasion tactics and the development of multidrug-resistant strains. This prompted the need to explore novel therapeutic approaches as an alternative to antibiotics. The current study aimed to develop a broad-spectrum, multi-epitope vaccine to control bacterial infections and reduce the burden on healthcare systems. A computational framework was designed to filter the immunogenic potent vaccine candidate. This framework consists of pan-genomics, subtractive proteomics, and immunoinformatics approaches to prioritize vaccine candidates. A total of 12,285 core proteins were obtained using a pan-genome analysis of all strains. The screening of the core proteins resulted in the selection of only two proteins for the next epitope prediction phase. Eleven B-cell derived T-cell epitopes were selected that met the criteria of different immunoinformatics approaches such as allergenicity, antigenicity, immunogenicity, and toxicity. A vaccine construct was formulated using EAAAK and GPGPG linkers and a cholera toxin B subunit. This formulated vaccine construct was further used for downward analysis. The vaccine was loop refined and improved for structure stability through disulfide engineering. For an efficient expression, the codons were optimized as per the usage pattern of the *E coli* (K12) expression system. The top three refined docked complexes of the vaccine that docked with the MHC-I, MHC-II, and TLR-4 receptors were selected, which proved the best binding potential of the vaccine with immune receptors; this was followed by molecular dynamic simulations. The results indicate the best intermolecular bonding between immune receptors and vaccine epitopes and that they are exposed to the host’s immune system. Finally, the binding energies were calculated to confirm the binding stability of the docked complexes. This work aimed to provide a manageable list of immunogenic and antigenic epitopes that could be used as potent vaccine candidates for experimental in vivo and in vitro studies.

## 1. Introduction

*Staphylococcus hominis* is a Gram-positive nosocomial pathogen and a member of the *staphylococcus* genus, consisting of spherical cells [1]. Despite being a harmless commensal to human and animal skin, it potentially causes bloodstream infections in immunocompromised patients. Among coagulase-negative staphylococci (CONs), *S. homins* is one of the three most identified isolates from the blood of hospitalized patients [2]. Studies have reported that among staphylococcal infections, 15% were caused by *S. hominis* and were also associated with several other diseases, including peritonitis, osteomyelitis, bone and joint infection, cancer, and bacterial meningitis [3,4,5,6,7]. *S. hominis* is a multidrug-resistant bacterium that is resistant to a class of antibiotics known as lactams (methicillin, tetracycline, erythromycin, novobiocin, and oxacillin). These were once effective against the bacterium, but they have now been replaced by vancomycin. The *S. hominis* sub-species novobiosepticus now shows some resistance to vancomycin. The staphylococcal cassette chromosome (SCC) is a mobile genetics element that has genes coded for antibiotic resistance proteins [8]. Apart from the classical immunization methods and Pasteur’s vaccinology, the genome sequencing of various microbes revolutionized vaccine development, allowing scientists to design vaccines using various computational approaches, tools, and software.

Reverse vaccinology (RV), a new computational approach, was used to develop the first computationally designed vaccine against *Meningococcus B* (MenB), a pathogen that was reported to cause 50% of meningococcal meningitis worldwide. [9,10]. RV is an effective approach for screening the immunogenic targets of a pathogen’s genome. The pan-genome is merged with the RV termed pan-genomic-based RV (PGRV) for the purpose of screening the intraspecific diversity of the bacteria and its broad-spectrum applications. Because of the emergence of antibiotic resistance (AR) in *S. hominis* and the lack of a licensed vaccine against this specific bacterium, the risk of HAIs is exponentially increasing.

Based on this, the current study was designed to explore novel therapeutics of the multi-epitope vaccine construct via the application of a computational framework that consists of subtractive proteomics, pan-genome, and RV. To prioritize promising vaccine candidates, we used the bacterial pan-genome of *S. hominis* and its subspecies, as well as immunoinformatics and RV-based approaches. An online server, the Immune Epitope Database (IEDB), was used to find B and T cell epitopes. All of the selected epitope sequences were further screened computationally to determine their immunogenicity, toxicity, allergenicity, and solubility. Adjuvants and linkers were combined with these sequences to efficiently boost immune responses. The vaccine construct was then modelled using the 3Dpro program of the Scratch protein predictor web server. Docking and molecular dynamic simulations were used to predict the binding affinity of the designed vaccine with different immune receptors. The stability of molecules was predicted by estimating the complex’s binding free energies. Because the RV approach uses several in silico filters to select high probability proteins as vaccine candidates from the coding DNA of the organism, the findings of this study will aid in the fast and efficient development of a vaccine against *S. hominis* by applying experimental testing, including in vitro and in vivo studies.

## 2. Materials and Methods

The proteomic data of *S. hominis* were retrieved from the genome database of the National Center for Biotechnological Information (NCBI). The data were then filtered and prioritized by many immunoinformatics tools for the identification of potential vaccine candidates. This prioritization is based on the criteria given in the literature [11]. The schematic view is given in Figure 1.

### 2.1. Pre-Screening

In this phase, we screened for the proteins that were conserved among all strains of the bacteria. We conducted a pan-genome analysis and screened the core genome sequence. The redundant proteins are not potent immunological proteins because of their poor conservation across the genomes of strains [12]; they are not part of the core genome. The non-redundant proteins in the core sequence were predicted using the CD-HIT web server with a sequence identity threshold of 90%, keeping all values default [13]. The non-redundant proteins were subjected to a subcellular localization check. Proteins that are localized at the surface of a pathogen or secretome are the vital proteins for a vaccine as they are in frequent contact with the host and cause infections [14]. The antigenicity of these proteins can be easily recognized by the host’s immune system and trigger an immune response. Proteins localized at the inner and outer membrane, periplasmic spaces, and secretory proteins were selected for the next phase. Proteins with multiple and unknown localizations were discarded [15].

### 2.2. Vaccine Candidate Prioritization

The pathogenic secretome and exoproteome were further filtered to find the proteins associated with pathogenesis. The screened proteins at this stage were inputted into the BLASTP software against the Virulent Factor Database (VFDB) for the screening of virulent proteins with a sequence identity ≥30% and bit score ≥100 [16]. The virulent proteins were further processed for physiochemical properties. The physiochemical characterization was performed through the ProtParam online tool. This tool predicts the computation of parameters such as instability index, molecular weight, estimated half-life, atomic composition, aliphatic index, theoretical PI, etc. [17]. The key factors to evaluate were the instability index and molecular weight. The proteins with instability index values of less than 40 and molecular weights of less than 100 kDa were considered the best choices for vaccine candidates. The next step was to examine the protein for transmembrane helices. Only proteins with 0 or 1 transmembrane helix were selected [18]. To eliminate the risk of inhibition of certain beneficial bacteria, the homology of proteins was checked against probiotics Lactobacillus species including *L. rhamnosus* (taxid: 47,715), *L.casei* (taxid: 1582), and *L.johnsonii* (taxid: 33,959) [19]. For this step, a BLASTP search was performed against them, and we only selected proteins that had no significant similarity [20].

### 2.3. Prediction of Immune Cells Epitopes

With respect to the IEDB data, the Bepipred linear epitope prediction method with a cut-off value of 0.5 was utilized to predict linear B-cell epitopes [21]. The B-cell epitopes were subsequently used for T-cell antigenic determinants using the IEDB MHC-I and MHC-II epitope predictors. A comparative analysis was performed to select subsequences that bound to both MHC-I and MHC-II alleles [22]. The method employed for this prediction was IEDB 2.22, where the peptides are sorted based on a percentile score, with the lower percentile score having a higher binding affinity [23].

### 2.4. Epitopes Prioritization

In this phase, the selected epitopes were subjected to an epitopes prioritization phase for the filtration of antigenic, non-allergenic, non-toxic, and soluble epitopes [24]. The antigenicity was checked using the VaxiJen v.2.0 server [25]. The allergenic epitopes were checked through the AllerTOP v.2.0 server (https://www.ddg-pharmfac.net/AllerTOP/method.html, accessed on 1 June 2022) [26]. The toxicity was predicted by using a bioinformatics tool called ToxinPred (http://crdd.osdd.net/raghava/toxinpred/, accessed on 1 June 2022) [27]. The water-soluble epitopes were evaluated using the Innovagen online tool (solubility peptides calculator) (https://pepcalc.com/peptide-solubility-calculator.php, accessed on 1 June 2022). For the above antigenicity analysis, a 0.4 threshold value was used; epitopes having a value of ≥0.4 were considered antigenic epitopes.

### 2.5. Multi-Epitope Peptide Designing

Peptide vaccines have weak immunogenicity, which can be overcome by using immunodominant epitopes (adjuvants) [28]. The multi-epitope peptide vaccine has series of overlapping epitope protein sequences that can be assembled and combined using GPGPG linkers [29]. The 3D structure of the construct was simulated using the 3Dpro program from the Scratch Protein Predictor web server [30]. Furthermore, the structure was refined using the Galaxy Refine tool of the Galaxy web server [31].

### 2.6. Molecular Docking

Molecular docking plays a vital role in vaccine construct designing to interpret the binding affinity of the vaccine construct with the host’s immune receptors [32]. An effective immune response will be activated when the MEV efficiently binds to the host’s immune receptors. The molecular docking of the designed vaccine construct with immune receptors was performed using the web server PATCHDOCK, which allows for the docking of two interacting molecules based on the interrelation principles of the shapes [33]. A blind docking strategy was applied with the immune receptors, TLR4, MHC-I, and MHC-II molecules [34]. The input clustering for the RMSD and complex type were set as their defaults. The docked complexes were then refined using FireDock on the same web server. FireDock re-ranked the docking complex results by filtering out all clashes and molecular conformational errors [35]. The topmost selected complexes were then visualized using the UCSF Chimera 1.16 program.

### 2.7. In Silico Cloning and Codon Optimization

To obtain an efficient expression of the vaccine construct in the *E. coli* expression system, the sequence was reverse-translated and optimized for codon usage [36]. The vaccine sequence was reverse-translated and improved by utilizing the JCAT (Java Codon Adaptation Tool), which calculates the CAI (codon adaptation index) and GC content of the sequence. Ideally, the CAI should be one and the GC content should be around 30–70% [37]. The sequence was then cloned into the pET-28a (+) expression vector of *E. coli* (K12) using a software called SnapGene [38].

### 2.8. Disulfide Engineering

The MEV construct was engineered using a disulfide engineering approach for structure stability. The disulfide links were introduced into the structure using the bioinformatics tool Design2.0 [39].

### 2.9. Molecular Dynamic (MD) Simulations

An MD simulation is a computational framework to evaluate the structural dynamics of biomolecules at the atomic resolution with different environmental conditions. A software called Assisted Model Building and Energy Refinement (AMBER) v20 was used in our process [40]. This process has three phases: system preparation, preprocessing, and production. In the first phase, the top 3 docked complexes of the MEV were prepared using the antechamber program. The libraries and parameters were set. Afterward, the complexes were added to the solvation box (TIP3P, size 12 Å) by the leap program. For the neutralization of the system, counter ions were added. The energy of the system was minimized step-by-step during the preprocessing phase, including the energy minimization of hydrogen atoms, water box, and non-heavy atoms. The ensembles such as NVT (constant temperature, constant volume) and NPT (constant temperature, constant pressure) were used to increase the temperature and pressure, respectively. The hydrogen bond constraint was maintained using the SHAKE algorithm. The temperature was maintained using Langevin dynamics. For 1 ns, the system was allowed to equilibrate itself. In the production phase, the simulation was performed for 100 ns at a time scale of 2 fs using the Berendsen algorithm [41].

### 2.10. Binding Free Energies Calculation

The binding free energies of the top 3 docked complexes were calculated using the molecular mechanics/Poisson–Boltzmann surface area (MM/PBSA) and molecular mechanics/generalized Born surface area (MM/GBSA) approaches by employing the MMPBSA.py module of AMBER v20 [42]. A total of 1000 frames of the trajectory of simulation were selected to calculate the binding free energies. The main goal was to find the difference between the free energy of the solvated and non-solvated states of the complexes [43].

## 3. Results

### 3.1. Proteins Sequence Retrieval

The sequences of the following bacterial strains were retrieved from the NCBI, as mentioned in Table 1.

### 3.2. Pan-Genome Analysis

The development of metagenomics and next-generation sequencing technologies has led to the focus shifting on the analysis and study of multiple genomes of different bacterial strains altogether rather than a few genome comparisons. The concept of a pan-genome is the outcome of such a multi-genome analysis [44,45]. This analysis provides an overview of the horizontal gene transfer and insights into the evolution of species [46]; it provides a detailed summary of the genomic diversity of the dataset, determining the core, accessory, and unique gene pool of the bacterial species [47]. Using the BPGA pipeline, we analyzed the pan-genome of all strains of *S. hominis*. This approach categorizes all the protein sequences based on the genetic conservation among them [48]. The core genome contains the sequences that are conserved among all the strains. The parts of the genome that are common in few strains are called accessory genes. Unique genes found only in a single strain are called singletons [49]. The output of the BPGA analysis is mentioned in Table 2.

### 3.3. CD-HIT Analysis

The core sequences were used further down the line. The core sequences are the pool of redundant and non-redundant proteins. The CD-HIT method was used to identify redundant and non-redundant proteins [50]; CD-HIT is a super-fast protein sequence clustering web server that can analyze millions of proteins and group together similar proteins into clusters based on sequence similarity. The core genome of *S. hominis* consists of 12,285 protein sequences that were analyzed with a sequence identity cut-off value of 90%. The proteins that are similar by up to 90% were discarded. Because of this filter, the redundant sequences (paralogous) were removed, leaving only the non-redundant sequences (orthologous), which included 1824 protein sequences [51].

### 3.4. Sub-Cellular Localization

In this phase, we localized the proteins of the orthologous sequence of *S. hominis*. This is the most important step in selecting effective vaccine candidates. Proteins found in the inner and outer membranes, periplasmic spaces, and secretory proteins are promising candidates for vaccine construct development [52]. These proteins play vital roles in the pathogenesis, invasion, and colonization of bacteria in a host’s cells. Furthermore, the proteins are all surface proteins, and the antigenic epitopes of these proteins can efficiently be recognized by the host immune system and provoke innate immune responses [53]. Out of 1824 non-redundant sequences, only 22 extracellular protein sequences were opted for further processing, as presented by the different proteins and numbers in Figure 2. The online tool PSORTb was used in the categorization of the protein localities. This tool is broadly used for the determination of the sub-cellular localization of proteins [54].

### 3.5. Virulence Check

Virulence is the pathogen’s ability to cause an infection in a host cell and initiate immune responses [55]. The core redundant exoproteome/secretome was checked for virulence. This filtration of proteins was performed through the VFDB database, with specific criteria of selection as mentioned in the methodology of [56]. A BLASTP search was performed to shortlist the proteins. Out of the 22 non-redundant core sequences, only six (27.3%) protein sequences were found to be virulent, and they are listed in Table 3.

### 3.6. Transmembrane Helices and Physiochemical Properties Analysis

Different physiochemical properties and transmembrane helices were evaluated to filter out the stable virulent proteins. The proteins with transmembrane helices of zero or one were selected. Such choices were made because of the failure of poor protein expression in in vitro systems such as *E. coli*. Having more than two transmembrane helices makes it difficult to conduct studies and cloning [57]. In the physiochemical properties, the prime factor to evaluate is molecular weight. The isolation and purification of proteins with low molecular weight are easy to perform. The other factor to analyze is the instability index. The instability index computes the stability of proteins by analyzing the disulfide bonds in proteomic sequences. Out of the six virulent proteins, three (50%) proteins were screened to be stable proteins. Table 4 summarizes the output results of the physiochemical properties and transmembrane helices. A threshold value of 40 was used for the instability index. Proteins having an instability index value greater than 40 were considered unstable and discarded from the study. Proteins having an instability index of less than 40 were considered stable and subjected to further analysis.

### 3.7. Homology Check against Probiotics

In the NCBI database, a BLASTP search was performed against probiotic lactobacillus bacteria species. These bacteria are commensals in the human gut and help to prevent diarrhea, improve gut health, lipid metabolism, and modulate immune and inflammatory responses [58]. Any type of homology between the probiotic and pathogen proteomic sequence could disturb gastrointestinal health and worsen the situation. In this filtration, only two proteins were found to have no significant sequence similarity [59]. For the comparison purpose, the non-homologous proteins must have ≤30% of sequence identity with the probiotic bacteria and must have an E-value of 10−4.

### 3.8. B and T Cells Epitope Prediction

Specificity in killing pathogens is the foundation of adaptive immunity [60]. B and T cells are the fundamental units of adaptive immunity. These cells recognize and kill the pathogens by a series of chemical reactions and store the data in their memory cells to prevent the same infection in the future [61]. The process of memorizing the pathogen’s identity and immunological data becomes the fundamental principle of vaccination. The major cells involved in adaptive/acquired immunity are the T and B lymphocytes [62]. They initiate two types of responses: humoral and cell-mediated immune responses against specific pathogens. The filtered two proteins were used to predict the B cells’ epitopes, and then these epitopes were subsequently used for the T cell epitope mapping. B cell epitope prediction is important because these epitopes could activate responses such as agglutination, neutralization, opsonization, complement system, and cell-mediated cytotoxicity administrated by antibody lymphocytes (natural killer cells, eosinophil, etc.) which can destroy the invader pathogen [63]. These epitopes were further used for T cell epitope mapping, where important sites for the MHC class I and II molecules were predicted. The MHC-I molecules are present on the surface of all nucleated body cells and present the pathogenic proteins to the cytotoxic T cells. In the same way, MHC-II molecules are expressed on the surface of special antigen-presenting cells such as dendritic cells, monocytes, and B lymphocytes [64]. The set of alleles used for prediction of MHC-II epitopes were HLA-DRB1*01:01, HLA-DRB1*03:*04:01, HLA-DRB101, HLA-DRB1 *04:05, HLA-DRB1*07:01, HLA-DQA1*03:01/DQB1*03:02, HLADQA1*03:01/DQB1*03:02, HLADQA1 *01:02/DQB1*06:02, HLADPA1*02:01/DPB1*01:01, HLADPA1*01:03/DPB1*04:01, HLADPA1 *03:01/DPB1*04:02, HLADPA1*02:01/DPB1*05:01, and HLADPA1*02:01/DPB1*14:01. For MHC-I epitopes, the alleles used were HLA-A*01:01, HLA-A*01:01, HLA-A*02:01, HLA-A*02:01, HLA-A*02:03, LA-A*02:03, HLA-A*02:06, HLA-A*02:06, HLA-A*03:01, HLA-A*03:01, HLA-A*11:01, HLA-A*11:01, HLA-A*23:01, HLA-A*23:01, HLA-A*24:02, HLA-A*24:02, HLA-A*26:01, HLA-A*26:01, HLA-A*30:01, HLA-A*30:01, HLA-A*30:02, HLA-A*30:02, HLA-A*31:01, HLA-A*31:01, HLA-A*32:01, HLA-A*32:01, HLA-A*33:01, HLA-A*33:01, HLA-A*68:01, HLA-A*68:01, HLA-A*68:02, HLA-A*68:02, HLA-B*07:02, HLA-B*07:02, HLA-B*08:01, HLA-B*08:01, HLA-B*15:01, HLA-B*15:01, HLA-B*35:01, HLA-B*35:01, HLA-B*40:01, HLA-B*40:01, HLA-B*44:02, HLA-B*44:02, HLA-B*44:03, HLA-B*44:03, HLA-B*51:01, HLA-B*51:01, HLA-B*53:01, HLA-B*53:01, HLA-B*57:01, HLA-B*57:01, HLA-B*58:01, and HLA-B*58:01. Only these epitopes were selected that are common in both MHC-I and MHC-II molecules. During the prediction, it was preferred to select 9-mer epitopes due to their chemical stability (doi:10.1111/ajt.13598 accessed on 10 June 2022).

This displaying of the pathogenic proteins leads to the full-force production of antibodies [65]. The predicted B and T cells’ epitopes are presented in Table 5 and Table 6, respectively.

### 3.9. Epitope Prioritization

Using different immunoinformatics approaches to prioritize the epitope selection, we determined the successful vaccine candidate to be water-soluble, non-toxic, non-allergenic, and antigenic proteins [66]. The antigenicity checks measure the ability of the antigen to bind to antibodies and other lymphocytic products [67]. The non-toxic, non-allergenic, water soluble, and antigenic epitopes are presented in Figure 3.

### 3.10. Multi-Epitope Vaccine Construct Processing

As the MEV is the combination of many epitopes, these epitope domains were combined using GPGPG linkers [16]. Because protease enzymes can easily degrade the antigenic peptides and epitopes, the B cell receptor (BCR) and T cell receptor (TCR) cannot recognize them and thus have a poor immunogenic effect [68]. In order to avoid this, the intermolecular adjuvants were used to boost the immune responses of the MEV and enable efficient transmission inside the body [69]. This adjuvant molecule was linked with the epitopes constructed with the help of the EAAK linker. The cholera toxin B subunit adjuvant was used in the process [70]. The 3D structure of the construct was modeled using the 3Dpro program of Scratch protein predictor. The structure visualized using Chimera 1.16 is presented in Figure 4, whereas the schematic representation is mentioned in Figure 5.

### 3.11. Codon Optimization

The codons are the complimentary nucleotide triplets of the genetic codes encoded in DNA [71]. They decide which amino acid should be added to the growing protein chain in protein synthesis. There are 64 codons for 20 different types of amino acids. Different organisms use different codons for the same amino acid [72]. To obtain the maximum expression in the host cells, we must optimize the codons accordingly. The protein sequence of the MEV was reverse-translated into DNA using a codon optimization technique, which optimizes the given sequence according to the host’s codon usage pattern. In this case, we selected *E. coli* (K12) as a host for our sequence expression [73]. The GC content of the sequence is 52%, and the codon adaptation index is 0.95%. These values are considered ideal for the expression process. At last, the optimized MEV was cloned into the pET-28a (+) expression vector. The cloned vector is presented in the Figure 6.

### 3.12. Disulfide Engineering

In protein engineering, improving the stability of proteins is one of the main goals. The best logical approach is to improve the stabilizing molecular interactions that are naturally found in proteins [74]. Disulfide bridges are covalent bonds that provide considerable structural stability. Disulfide engineering is a method of directing disulfide bonds into the vaccine construct to make it structurally stable. Many residues are enzyme degradable, so they are replaced with cysteine residues in the vaccine construct [75]. The mutated residues are tabulated in Table 7 while the mutant and original structure of the designed vaccine is mentioned in Figure 7.

### 3.13. Molecular Docking

Molecular docking analysis approaches were used to elucidate the interaction between the vaccine construct and the host’s innate and adaptive immune cells [76]. To provoke immune responses, the MEV should have good interactions with the host’s immune cells receptors such as TLR4, MHC-I and MHC-II. For this purpose, a blind docking strategy was used to predict the interaction of the MEV with the host’s immune receptors [77]. The results are tabulated in Table 8, Table 9 and Table 10. The server generated docking solutions, which were ranked on their binding energy score. The docked complexes having the best binding energy score were considered to be stable complexes. In each case, 20 solutions were generated, and the one with the highest binding energy score was selected. For instance, Solution 1 was chosen in all three receptors, and the vaccine-MHC-I complex had a best score of 19,690; the vaccine-MHC-II had a score of 19,030, and vaccine-TLR-4 had a score of 20,864.

### 3.14. Docking Refinement

The PATCHDOCK results were further refined using FireDock. The top 10 docked complexes were subjected to refinement [78]. FireDock refinement ranks the results with the lowest binding energy complex at the top and proceeds downward. The docked complexes having the lowest binding energies were selected for the simulation [79]. The FireDock results are tabulated in Table 11, Table 12 and Table 13, and the docked confirmation is presented in Figure 8.

### 3.15. MD Simulation

An MD simulation is a method for understanding molecular behavior and properties at the atomic level; it provides useful information about the structure, interaction energies, and movement of atoms (dynamics), which complements the experimental data [80]. In this process, the interaction of two molecules is evaluated at different conditions of temperature and pressure for a short period of time by using different algorithms and ensembles. The RMSD value was calculated based on the alpha carbon atom. The top three complexes were analyzed. The graph shows a steady line, although the vaccine and TLR4 complex show some deviations, seen by the line fluctuating [81] around the value 12–14Å. The main goal here was to investigate the vaccine binding interaction with the host’s immune cells and initiate immune responses. A graph of the RMSD is presented in Figure 9.

### 3.16. Binding Free Energies Calculations

The binding free energies of the top three docked complexes were calculated using the MM/GBSA approach [82]. The total binding free energies of the TLR4, MHC-I and MHC-II complexes were −74.47 kcal/mol, −62.25 kcal/mol, and −71.95 kcal/mol, respectively. The estimated values are tabulated in Table 14. As can be seen in the table, both the van der Waals and electrostatic energies majorly contributed to the intermolecular binding and showed a stable binding conformation.

## 4. Discussion

The current study is an in silico based-model of a multi-epitope vaccine against *S. hominis*. We used a computational framework to evaluate the core proteomes of *S. hominis* for non-homologous, antigenic, and highly conserved proteomes among all the strains of *S. hominins.* These proteins were further used to screen antigenic, non-allergenic, non-toxic, and water-soluble epitopes for B and T cell alleles. Further analysis (such as analyzing the docking, MD simulation, and binding energies calculation results) revealed great immunogenic effects and stable molecular configurations [83].

In recent history, the exponential growth of antimicrobial resistance has been observed because of the overuse of antibiotics that have often deviated from prescription. Once the strain emerges with antibiotic resistance, it can quickly spread and acquire resistance to other classes of drugs as well. These multidrug-resistant bacteria limit the choice of choosing specific antibiotics, increasing the mortality and morbidity rates. It is estimated that by 2050, 10 million lives a year may be lost to AMR, exceeding the 8.2 million lives a year currently lost to cancer. To put this number into perspective, at least 700,000 people die of resistant infections every year worldwide, more than the combined number of deaths caused by tetanus, cholera, and measles [84]. Furthermore, it is a big loss to the economy. It has been estimated that, if the AMR trend continues, the cumulative loss to world economies might be as high as USD 100 trillion by 2050 [85]. The development of new drugs cannot keep pace with the increasing risk of antimicrobial resistance. However, the immunization of individuals through vaccination can affect this both directly and indirectly [86]. As the development of vaccines greatly reduce the intake of antibiotics, this in turn decreases the chances of new antimicrobial-resistant strains emerging. Traditional vaccinology approaches have several defects, i.e., they generate inaccurate immune responses, and have lengthy processing times, high cost, less specificity, less safety, hypersensitivity, and less stability [87]. Computational vaccine designing strategies are exponentially growing, mainly because of the huge growth of genomic data that has allowed scientists to move beyond Pasteur’s rule of vaccinology and instead use computational approaches, tools, and software to design vaccines without the need for any wet-lab techniques. A previous study conducted by Aldakheel et al. designed an MEV using proteome-wide mapping and reverse vaccinology and used the same computational framework to model highly antigenic and potent MEV targets against *Clostridium perfringens* [70]. The same computational approach was used by Al-Megrin et al. They designed a novel MEV against *Staphylococcus auricularis* using immunoinformatics and biophysics approaches. They prioritized the vaccine candidate on the tight criteria provided in the literature. The vaccine candidates are non-homologic, conserved, and immunogenic. The epitopes derived from these proteins were also immunogenic, non-allergic, non-toxic, and water-soluble. These epitopes were derived from B and T cell alleles. This MEV construct has a higher molecular stability and efficient immune response, according to the simulation and binding energies calculations [70].

## 5. Conclusions

Our work presents an in silico design of a broad-spectrum multi-epitope vaccine against *S. hominis*. The vaccine construct is composed of antigenic, non-allergic, non-toxic multi-epitopes extracted from highly virulent proteins that are part of a pathogen’s core exoproteome [88]. Many immunoinformatics, biophysical, and subtractive proteomic techniques were used in the process [89]. During the selection of a vaccine candidate for vaccine development, tough selection criteria were used, such as that proteins should be from the core proteome, should be present on cell surface (or be excretory), should be non-homologous to the host, and should be probiotics [90]. The epitopes that were prioritized were those with non-allergenic, non-toxic, antigenic, and water-soluble properties that had a higher binding affinity to B and T cell alleles. The results of the MD simulation and binding energies calculations elaborated the stable molecular configuration and minimize system energies. Although the pan-genome-based reverse vaccinology is an effective method to develop a multi-epitope vaccine, the potency of the vaccine should be studied and confirmed by in vivo and in vitro immunological methods [91].

## Figures and Tables

**Figure 1 vaccines-10-01729-f001:**
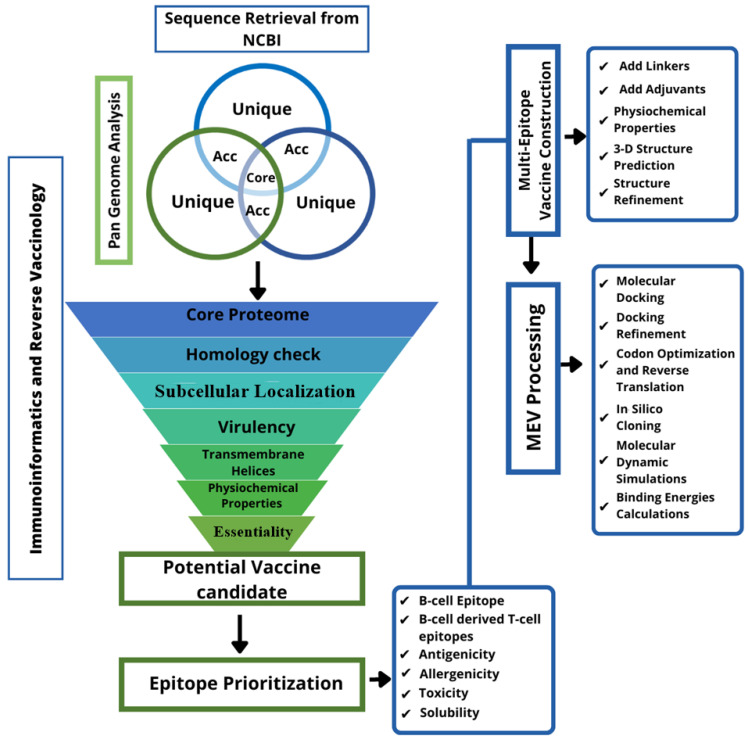
Schematic view of a designed computational framework for screening a broad-spectrum vaccine against *S. hominis* in the reference proteome followed by epitope prioritization, multi-epitopes vaccine (MEV) construction, and MEV processing.

**Figure 2 vaccines-10-01729-f002:**
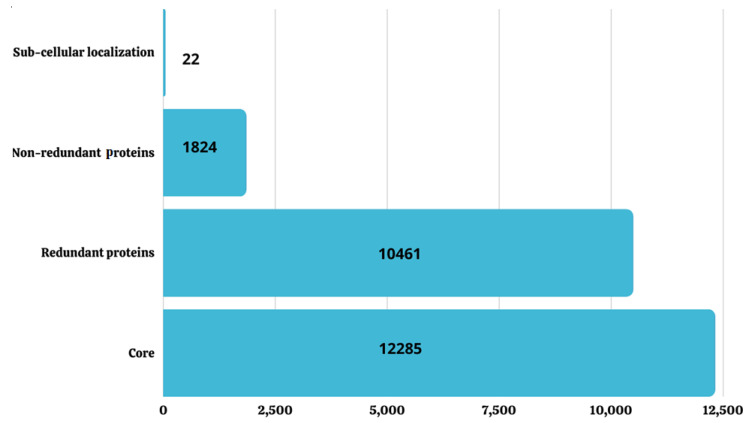
The overview of screening core proteins from different steps.

**Figure 3 vaccines-10-01729-f003:**
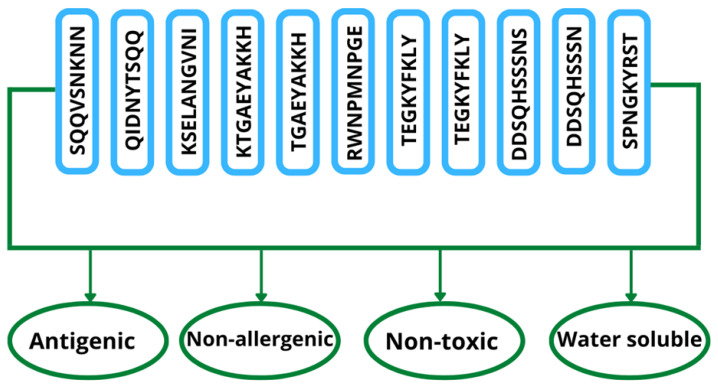
Number of 11 shortlisted epitopes that are non-toxic, non-allergenic, antigenic, and water-soluble.

**Figure 4 vaccines-10-01729-f004:**
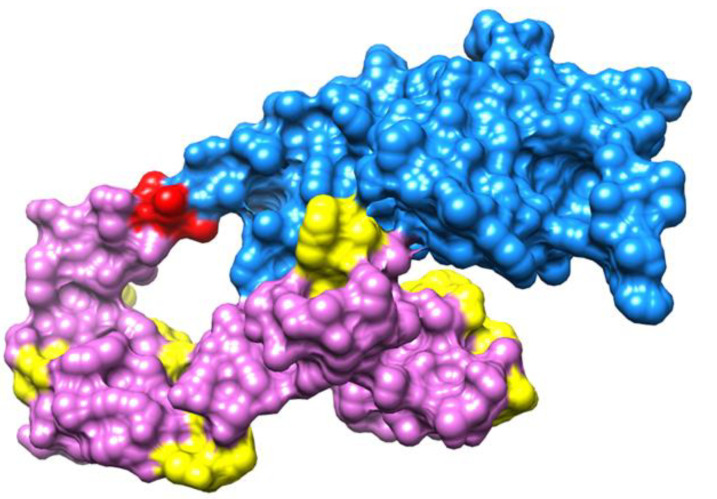
A visualization of the MEV construct. The yellow color represents GPGPG linkers, blue color represents the cholera toxin B subunit, and red (EAAAK linkers) and orchid colors represent the epitopes.

**Figure 5 vaccines-10-01729-f005:**
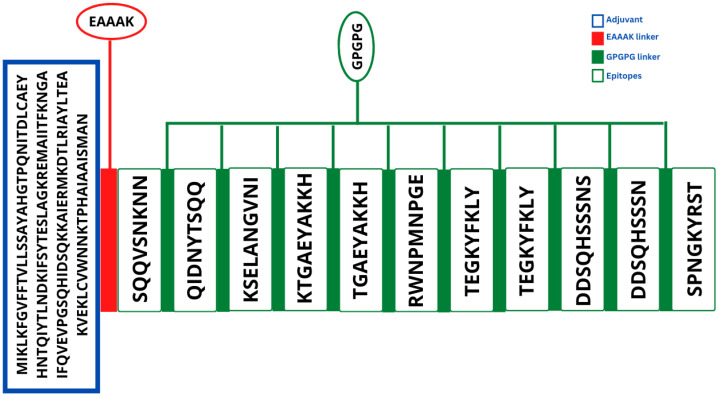
The schematic description of the vaccine construct. The blue colored text box shows the cholera toxin B subunit (adjuvant) sequence. The red colored text box depicts the EAAAK linker. The white colored boxes depict the selected epitopes that are joined by GPGPG linkers, as indicated by the green colored textboxes.

**Figure 6 vaccines-10-01729-f006:**
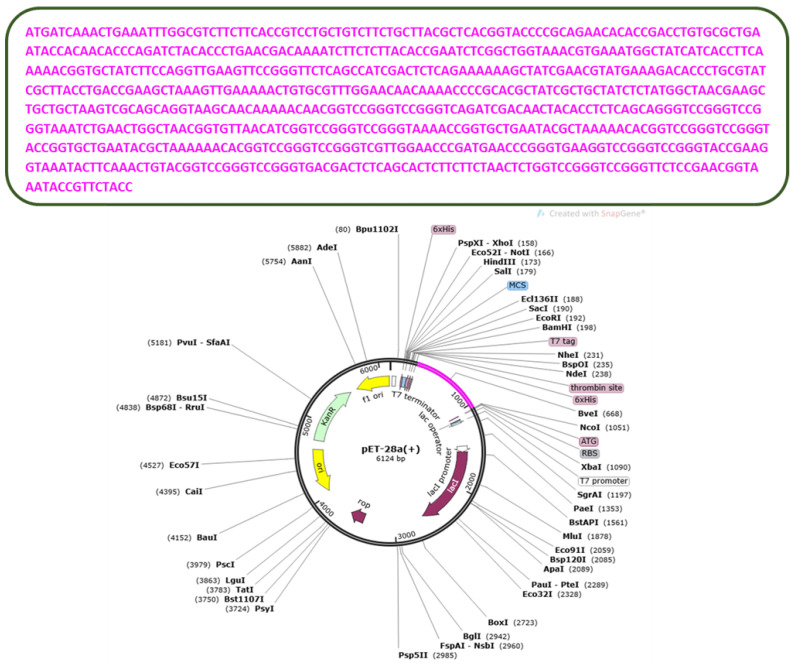
The cloned pET 28a (+) Vector.

**Figure 7 vaccines-10-01729-f007:**
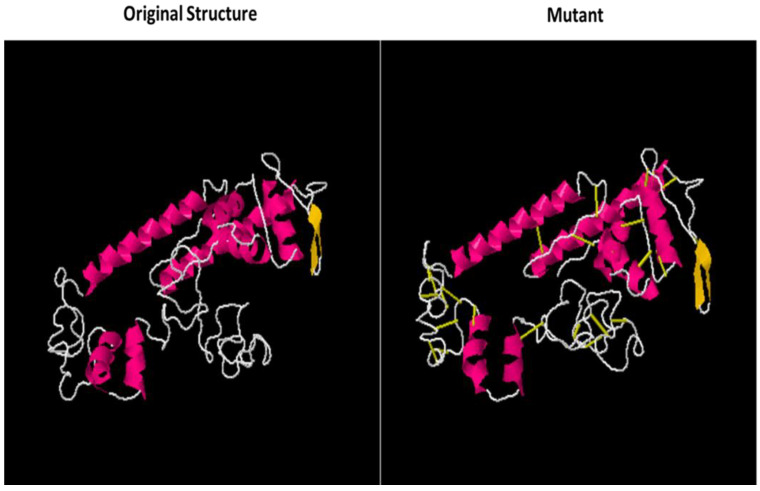
Original and mutant structure of the MEV construct. The yellowish highlighted areas in the mutant shows the introduction of the disulfide bonds into the structure.

**Figure 8 vaccines-10-01729-f008:**
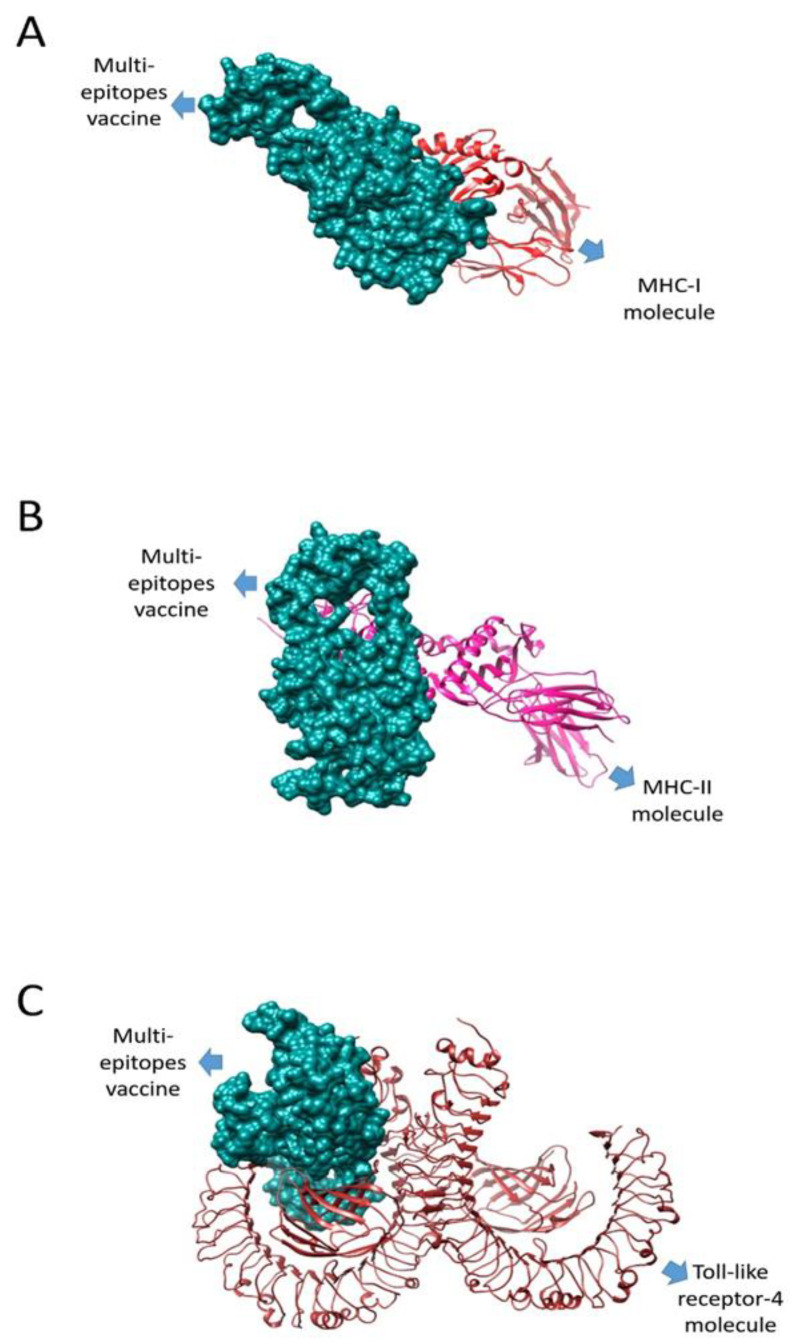
Intermolecular docked complexes of the vaccine with the MHC-I (**A**), MHC-II (**B**), and TLR-4 (**C**) molecules.

**Figure 9 vaccines-10-01729-f009:**
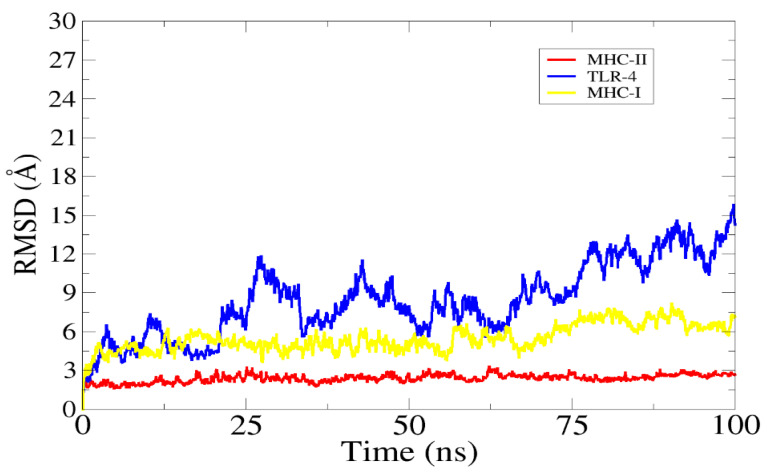
The root-mean-square deviation (RMSD) analysis graph.

**Table 1 vaccines-10-01729-t001:** Bacterial strains used in the process.

Organism Name	Strain	Size	GC%
*S. hominis*	FDAARGOS_575	2.25743	31.6637
	FDAARGOS_746	2.37219	31.6522
	19A	2.28122	31.6534
	S34-1	2.25454	31.4135
	FDAARGOS_747	2.24212	31.5385
	FDAARGOS_762	2.25551	31.529
	K1	2.25341	31.4

**Table 2 vaccines-10-01729-t002:** Details of the pan-genome analysis. ACC (Accessory genes).

Fasta Files	Core Genes	ACC Genes	Unique Genes	Exclusively Absent Genes
19A protein.faa	1755	211	101	13
FDAARGOS_747.protein.faa	1755	250	51	5
FDAARGOS_762.protein.faa	1755	232	49	9
FDAARGOS_protein.faa	1755	290	160	8
FDAARGOS_proteins.faa	1755	281	53	7
K1.proteins.faa	1755	237	74	58
S34-1.proteins. faa	1755	235	58	16

**Table 3 vaccines-10-01729-t003:** List of virulent proteins.

Proteins	VFDB	
Extracellular	Bit Score	Sequence Identity
>core/1166/1/Org1_Gene1302	211	49%
>core/1180/1/Org1_Gene1448	169	45%
>core/1802/1/Org1_Gene1985	287	70%
>core/99/2/Org2_Gene1570	132	44%
>core/16/4/Org4_Gene1407	1569	58%

**Table 4 vaccines-10-01729-t004:** Physiochemical properties and transmembrane helices analysis of selected proteins.

Extracellular Proteins	Transmembrane Helices	No. of Amino Acids	Molecular Weight	Theoretical PI	Instability Index	Aliphatic Index
>core/1166/1/Org1_Gene1302	1	282	32.28	8.37	17.03	76.49
>core/1180/1/Org1_Gene1448	0	321	56.32	10.51	42.52	85.39
>core/1679/1/Org1_Gene1002	0	200	22.73	4.99	30.44	81.45
>core/1802/1/Org1_Gene1985	1	181	20.81	9	20.19	74.75
>core/99/2/Org2_Gene1570	0	385	58.60	8.65	45.67	84.52
>core/16/4/Org4_Gene1407	0	402	63.05	9.32	44.64	85.39

**Table 5 vaccines-10-01729-t005:** Predicted B Cell epitopes.

B Cell Epitopes	B Cells Peptides
>core/1166/1/Org1_Gene1302 N-acetylglucosaminidase	QIFFKKVNEVEKVQHVNVTLDKAAAKQIDNYTSQQVSNKNNNAW RDASASEIKGAMDSSKFIDDDKQKYQFLDLSKY QGIDKNRIKRMLFDRPTLLKHTD
	KSELANGVNIDGKK
	EDPIKTGAEYAKKHGWDT
	SHDDQNTLYSMRWNPMNPGEH
	KTEGKYFKLYVYKDDQ
>core/1802/1/Org1_Gene1985 Thermonuclease family protein	HTGPFKDDSQHSSSNSTQIELKGK
	TVKPNTPVQPY
	LAREKYFSPNGKYRST

**Table 6 vaccines-10-01729-t006:** Predicted MHC-I and MHC-II epitopes (T cell epitopes) based on percentile rank.

T Cell Epitopes (MHC-I and MHC-II)			
MHC-II	Percentile Score	MHC-I	Percentile Score
KNRIKRMLFDRPTLL	0.21	MLFDRPTLL	0.01
		KNRIKRMLF	0.94
QKYQFLDLSKYQGID	6.1	YQFLDLSKY	0.01
		DLSKYQGID	71
		QKYQFLDLSK	4.5
IKGAMDSSKFIDDDK	6.2	KGAMDSSKF	1.1
		SSKFIDDDK	1.8
		GAMDSSKFI	1.5
NAWRDASASEIKGAM	4.6	SASEIKGAM	0.23
		NAWRDASAS	4.9
QIDNYTSQQVSNKNN	22	YTSQQVSNK	0.07
		SQQVSNKNN	17
		QIDNYTSQQ	2.8
VQHVNVTLDKAAAKQ	3.2	VTLDKAAAK	0.02
		VQHVNVTLDK	1.8
		VQHVNVTLDK	2.2
QIFFKKVNEVEKVQH	7.5	IFFKKVNEV	0.2
		KVNEVEKVQH	1.1
		QIFFKKVNEV	0.83
KSELANGVNID	7.2	SELANGVNI	0.14
		SELANGVNID	4.9
		KSELANGVNI	1.1
SELANGVNIDGKK	28	SELANGVNI	0.14
		NGVNIDGKK	3.2
DPIKTGAEYAKKH	15	DPIKTGAEY	0.02
		KTGAEYAKKH	3.3
TGAEYAKKHGWDT	31	AEYAKKHGW	0.01
		YAKKHGWDT	2.1
		TGAEYAKKH	4.4
NTLYSMRWNPMNPGE	3.3	SMRWNPMNP	1.6
		RWNPMNPGE	3.2
		NTLYSMRWNP	11
SHDDQNTLYSMRWNP	36	DQNTLYSMR	0.15
		TLYSMRWNP	3.5
		SHDDQNTLY	0.23
KTEGKYFKLYVYKDD	3.5	TEGKYFKLY	0.01
		YFKLYVYKDD	48
TEGKYFKLYVYKDDQ	4.3	TEGKYFKLY	0.01
		KLYVYKDDQ	15
DDSQHSSSNSTQIEL	14	HSSSNSTQI	0.63
		SSNSTQIEL	0.76
		DDSQHSSSNS	32
QHSSSNSTQIELKGK	18	SSNSTQIELK	0.06
		STQIELKGK	0.18
		QHSSSNSTQI	4.5
HTGPFKDDSQHSSSN	19	GPFKDDSQH	0.84
		DDSQHSSSN	23
		HTGPFKDDSQ	13
TVKPNTPVQPY	6	KPNTPVQPY	0.05
		TVKPNTPVQ	0.74
LAREKYFSPNGKYRS	4.4	KYFSPNGKY	0.01
		YFSPNGKYRS	3.8
		LAREKYFSP	1.1
AREKYFSPNGKYRST	4.5	KYFSPNGKY	0.01
		SPNGKYRST	0.2
		AREKYFSPNG	13

**Table 7 vaccines-10-01729-t007:** The pairs of amino acid residues. Chi3 values and energy.

Pair of Residues	Chi3	Energy
Leu4-Glu104	116.71	3.66
Phe10-Asn25	−88.2	5.92
Leu13-Asp28	109.19	3.15
Leu13-Asn35	80.16	3.35
Thr27-Tyr33	92.9	4.18
Thr36-Leu41	95.75	3.06
Ile38-Leu41	114.86	3.82
Thr49-Ala53	−99.93	4.39
Glu50-His78	−99.21	2.56
Met58-Ala67	−88.37	3.6
Pro74-Gln77	−103.37	2.03
Ala96-Lys102	−80.09	2.7
Ala96-Ala119	−106.52	1.15
Leu106-Trp109	93.83	3.1
Trp109-Lys112	79.05	0.99
Pro140-Thr149	−84.69	5.82
Gly143-Asp146	−78.71	2.98
Asn147-Ser150	78.25	2.72
Tyr148-Pro169	−99.43	5.6
Ser150-Gly153	99.22	0.78
Pro154-Gly168	−91.74	0.15
Gly157-Glu160	80.42	4.03
Ala162-Val165	−62.14	6.82
Gly170-Lys173	−106.59	3.22
Gly175-Ala179	120.42	9.13
Tyr192-Gly199	90.13	5.39
Phe221-Tyr224	101.86	2.6
Lys222-Gln233	−85.19	2.76
Gly225-Ser245	67.24	5.62
Gly227-Asp230	−95.1	2.68
Asp231-Pro243	−77.12	3.04

**Table 8 vaccines-10-01729-t008:** Top 20 docking scores of the vaccine and MHC-I, generated by the PATCHDOCK webserver.

Solution No.	Score	Area	Atomic Contact Energy	Transformation
1	19,690	2856.90	243.71	−0.03 0.30 −0.09 52.34 47.37 58.64
2	19,200	3027.60	270.89	0.75 0.42 0.94 57.44 55.30 16.13
3	18,192	3065.80	157.31	−3.09 −0.32 −0.24 62.27 28.26 48.61
4	17,630	3437.00	489.66	0.64 −0.12 −1.90 30.94 44.61 52.59
5	17,448	2402.00	332.04	−2.53 0.48 2.20 47.52 38.06 57.76
6	17,354	2529.80	356.97	−2.47 0.59 1.94 48.65 35.91 57.87
7	17,298	2026.20	211.10	−2.53 0.44 2.46 46.83 39.76 57.70
8	16,830	2582.40	421.25	−1.32 1.24 −0.69 16.14 44.00 37.89
9	16,788	2898.30	123.86	2.57 −0.79 −1.91 7.67 22.75 27.87
10	16,722	2468.90	80.82	0.36 −0.87 1.68 50.92 37.24 65.79
11	15,864	2500.60	400.30	2.90 1.04 −1.53 39.11 −3.56 38.58
12	15,804	2384.30	235.58	−0.21 0.41 −0.16 48.41 47.33 59.09
13	15,752	2633.60	211.87	1.34 −0.24 −3.02 25.94 15.16 18.01
14	15,648	2207.40	349.69	2.57 −0.63 −2.05 5.24 25.83 31.22
15	15,482	2210.30	339.58	0.90 0.63 2.53 10.72 23.61 32.81
16	15,454	2190.40	430.68	1.85 −0.76 −2.43 47.55 26.12 8.92
17	15,452	2184.50	448.16	−0.33 −0.92 0.49 51.82 12.75 19.05
18	15,436	2974.60	139.08	0.22 −0.06 −2.19 57.28 10.49 57.29
19	15,404	3022.70	446.40	−0.15 −0.29 0.67 13.83 38.25 47.47
20	15,390	3328.10	−35.46	2.70 −0.46 −1.70 46.00 55.26 63.86

**Table 9 vaccines-10-01729-t009:** Top 20 docking score of the vaccine and MHC-II, generated by the PATCHDOCK webserver.

Solution No.	Score	Area	Atomic Contact Energy	Transformation
1	19,030	2568.20	178.30	−2.52 0.46 0.61 103.23 100.92 −12.98
2	19,012	2616.00	174.36	3.12 −0.94 −3.01 118.54 31.94 −4.33
3	18,762	3157.30	339.60	−0.95 −1.48 −0.11 102.18 52.19 −15.79
4	17,928	2860.90	498.89	2.13 −0.68 2.42 115.91 101.86 1.25
5	17,924	3188.70	−118.16	−2.22 −0.86 −0.00 121.14 24.49 10.29
6	17,584	2759.50	−36.31	2.22 0.44 2.53 142.80 64.79 15.68
7	17,258	2463.10	365.25	2.36 1.27 −2.48 85.84 67.41 3.01
8	16,858	2782.20	189.39	−1.26 −0.73 −0.94 146.96 58.95 −5.46
9	16,806	2721.60	333.73	2.88 −0.76 −3.13 118.27 32.94 −6.90
10	16,788	2957.30	−44.01	−3.09 −1.10 −3.03 117.35 34.09 −6.06
11	16,616	2588.80	367.97	−2.23 0.84 0.94 72.25 63.69 10.94
12	16,606	2181.70	29.05	0.23 0.32 −2.44 94.64 88.42 −23.96
13	16,602	2646.70	223.64	0.41 0.79 0.94 122.16 92.19 −6.03
14	16,598	2646.80	289.18	−1.62 0.07 0.15 72.26 77.33 −6.13
15	16,542	2996.90	199.69	−0.30 −0.95 −2.67 82.55 81.78 14.62
16	16,534	2063.40	3.75	3.06 0.75 1.28 139.48 75.91 −15.37
17	16,532	2808.00	17.66	−1.36 −0.55 −0.95 149.27 54.23 −4.98
18	16,338	3494.90	200.24	2.42 −1.51 −3.10 99.91 52.71 −12.04
19	16,330	2461.90	116.63	0.88 0.49 2.74 89.56 69.78 −12.43
20	16,262	2662.00	81.94	1.13 0.51 2.68 89.25 66.06 −9.25

**Table 10 vaccines-10-01729-t010:** Top 20 docking scores of the vaccine and TLR-4, generated by the PATCHDOCK webserver.

Solution No.	Score	Area	Atomic Contact Energy	Transformation
1	20,864	2763.80	319.39	−2.98 0.85 1.45 −48.02 20.31 −23.51
2	19,780	2391.10	482.45	−2.50 −0.35 2.08 −16.75 42.27 −16.53
3	19,694	3221.60	338.75	0.84 −1.45 2.63 −0.69 18.65 −54.86
4	19,676	2551.00	315.79	−2.61 0.04 −1.92 9.24 32.35 −58.78
5	19,228	2566.00	228.48	−0.75 −0.01 −2.79 −33.15 41.12 −15.34
6	18,084	3343.40	246.14	0.32 −1.46 2.08 1.33 18.82 −51.37
7	18,058	4052.50	372.52	3.13 0.68 1.53 −43.62 21.45 −22.30
8	17,838	2669.80	222.59	−2.80 0.05 −1.83 −31.47 42.41 −10.22
9	17,764	2476.80	177.68	3.11 0.80 1.69 −50.10 18.45 −24.37
10	17,748	3277.60	−178.96	−1.03 −0.37 −1.37 17.61 33.74 −54.26
11	17,652	2927.20	7.50	0.63 0.97 0.47 −55.98 24.76 −26.27
12	17,566	3376.60	435.51	−0.34 0.53 −1.72 −58.27 11.71 −52.97
13	17,424	2759.50	95.73	−0.28 −0.02 0.43 −15.69 49.62 −18.02
14	17,340	3144.90	319.63	−2.21 1.12 −2.60 −48.43 35.44 −21.42
15	17,340	2747.30	93.26	2.49 0.83 −2.57 −57.86 24.72 −0.77
16	17,322	2663.00	183.48	−2.28 0.63 −0.33 −68.26 −18.01 4.10
17	17,226	2159.20	218.29	0.42 −0.86 1.59 35.96 −6.27 −64.19
18	17,218	2911.20	375.82	−1.15 −0.50 −0.61 16.58 15.77 −70.72
19	17,040	2249.90	421.54	0.38 −0.73 −1.75 −44.40 47.72 −11.97
20	17,006	2235.50	218.89	−2.60 0.40 1.70 −72.01 −1.85 5.10

**Table 11 vaccines-10-01729-t011:** Refined docked solution of the vaccine with MHC−I, generated by the FireDock webserver.

Solution Number	Global Energy	Attractive van der Waals	Repulsive van der Waals	Atomic Contact Energy	Hydrogen Bonds Energy
8	−4.84	−3.77	0.34	−1.23	−0.99
9	6.04	−0.55	0.00	0.99	0.00
5	9.41	−3.75	0.00	2.70	−0.32
4	9.55	−3.40	1.80	3.71	−0.46
2	10.22	−0.00	0.00	−0.00	0.00
7	12.30	−10.85	3.92	1.55	0.00
6	18.35	−3.40	1.18	2.00	0.00
10	24.67	−6.10	0.97	1.15	0.00
3	28.63	−36.11	16.03	10.96	−2.12
1	31.21	−14.77	29.21	5.98	−3.36

**Table 12 vaccines-10-01729-t012:** Refined docked solution of the vaccine with MHC-II, generated by the FireDock webserver.

Solution Number	Global Energy	Attractive van der Waals	Repulsive van der Waals	Atomic Contact Energy	Hydrogen Bonds Energy
9	2.55	−4.82	0.80	1.65	0.00
5	4.36	−1.93	1.65	−0.43	0.00
10	10.58	−3.28	1.32	3.89	−0.34
8	15.30	−32.77	30.61	13.93	−2.23
2	16.03	−7.26	2.03	5.34	−0.42
6	28.39	−9.07	24.19	6.31	−0.46
3	55.64	−8.79	10.16	13.06	0.00
4	444.29	−32.75	578.96	18.68	−4.13
1	922.72	−28.54	1153.02	19.35	−4.68
7	1040.22	−28.82	1310.23	15.62	−5.01

**Table 13 vaccines-10-01729-t013:** Refined docked solution of the vaccine with TLR-4, generated by the FireDock webserver.

Solution Number	Global Energy	Attractive van der Waals	Repulsive van der Waals	Atomic Contact Energy	Hydrogen Bonds Energy
8	−36.67	−32.83	33.92	2.84	−6.88
1	−3.76	−25.41	6.33	10.26	−2.29
4	2.59	−1.90	0.30	0.62	0.00
2	20.09	−49.11	52.92	16.08	−7.74
5	364.56	−27.61	498.29	1.45	−2.59
9	511.26	−63.30	728.16	0.06	−3.96
3	776.89	−38.99	1019.72	14.40	−4.25
6	887.29	−40.59	1162.81	10.78	−2.95
7	1292.51	−36.49	1632.36	20.70	−1.52
10	7376.28	−56.57	9336.35	−11.09	−6.70

**Table 14 vaccines-10-01729-t014:** Binding free energies calculations.

Energy Parameter	TLR-4–Vaccine Complex	MHC-I–Vaccine Complex	MHC-II–Vaccine Complex
MM/GBSA			
VDWAALS	−56.68	−51.38	−66.62
EEL	−33.65	−32.55	−25.87
Delta G gas	−90.33	−83.93	−92.49
Delta G solv	15.86	21.68	20.54
Delta Total	−74.47	−62.25	−71.95

Key: VDWAALS (van der Waals), EEL (electrostatic), Delta G gas (net gas phase energy), Delta G solv (net solvation energy), Delta Total (net energy of system).

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
