# Peer review of "Genome-Based Multi-Antigenic Epitopes Vaccine Construct Designing against *Staphylococcus hominis* Using Reverse Vaccinology and Biophysical Approaches"

_vaccines, 2022, doi:10.3390/vaccines10101729_

Round 1
Reviewer 1 Report
The author screened the core protein of Staphylococcus hominis and B-cell derived T-cell epitopes by means of pan genomics, subtractive proteomics and immunoinformatics approaches, and optimized the vaccine by using EAAAK and GPGPG linkers, cholera toxin B subunit to enhance the immunogenicity of multi epitope vaccine. At the same time, the docking complex was optimized by disulfide engineering to improve the vaccine stability. On this basis, the ability of the vaccine to interact with MHC-I, II and TLR-4 receptors was analyzed, and the binding potential of the vaccine to immune receptors was evaluated. In general, this work provides a solid method for vaccine screening and design, and provides a list of manageable immunogenicity and antigen epitopes, which is of high significance for vaccine design and development. However, the manuscript also has the following questions need to be further revised.
1. “Materials and Methods” 2.4: Selection criteria and results in epitope prioritization should be supplemented in detail.
2. The results of 3.6: the data of unstable proteins in Table 4 should also be displayed so that readers can better understand the screening principles through the data.
3. The comparison results and determination criteria of probiotics homology in 3.7 should be described in detail.
4. The result of 3.13 should describe what criteria are used to evaluate the interaction, so that which immune cell receptors are selected for subsequent evaluation, rather than simply listing the data.
5. The experimental results in 3.13-3.15 simulate the interaction between the vaccine and the selected immune cell receptors to prove the effectiveness of the vaccine, but I think these results alone cannot really prove the role of the vaccine. If possible, it is better to supplement the animals’ experimental results.
6. Some parts of the manuscript are not clearly described or too simple and should be supplemented to help readers better understand the article.
7. The title of Table 6 is written incorrectly with only one bracket.
8. The schematic diagram of Fig 5 is unclear, and there is no blue mark on the diagram.
The word of Table 4 “Transmembrane” and Table 13 “Solution” should be in the same line
Author Response
Response to Reviewer Comments
We thank the Referee for spending time and interest in our work and for helpful comments that will greatly improve the manuscript. We have checked all the general and specific comments provided by the Referee and have made all the necessary changes according to his indications. Please refer to yellow highlighted sections in the revised manuscript.
Reviewer # 2
Comments and Suggestions for Authors
The author screened the core protein of Staphylococcus hominis and B-cell derived T-cell epitopes by means of pan genomics, subtractive proteomics and immunoinformatics approaches, and optimized the vaccine by using EAAAK and GPGPG linkers, cholera toxin B subunit to enhance the immunogenicity of multi epitope vaccine. At the same time, the docking complex was optimized by disulfide engineering to improve the vaccine stability. On this basis, the ability of the vaccine to interact with MHC-I, II and TLR-4 receptors was analyzed, and the binding potential of the vaccine to immune receptors was evaluated. In general, this work provides a solid method for vaccine screening and design, and provides a list of manageable immunogenicity and antigen epitopes, which is of high significance for vaccine design and development. However, the manuscript also has the following questions need to be further revised.
- “Materials and Methods” 2.4: Selection criteria and results in epitope prioritization should be supplemented in detail.
Response: Thank you for valuable comment. The section is revisited in the revised manuscript as per reviewer comment.
- The results of 3.6: the data of unstable proteins in Table 4 should also be displayed so that readers can better understand the screening principles through the data.
Response: Thank you for valuable comment. The required data is added as per reviewer instructions.
- The comparison results and determination criteria of probiotics homology in 3.7 should be described in detail.
Response: Thank you for valuable comment. The required data is added as per reviewer instructions.
- The result of 3.13 should describe what criteria are used to evaluate the interaction, so that which immune cell receptors are selected for subsequent evaluation, rather than simply listing the data.
Response: Thank you for valuable comment. The required text is added as per reviewer instructions.
- The experimental results in 3.13-3.15 simulate the interaction between the vaccine and the selected immune cell receptors to prove the effectiveness of the vaccine, but I think these results alone cannot really prove the role of the vaccine. If possible, it is better to supplement the animals’ experimental results.
Response: We understand the reviewer concern. As the scope of the paper is computational, the main objective of the work is to provide theoretical vaccine model for experimentalists to check the designed vaccine immune protective efficacy against the said pathogen in vivo. Reverse Vaccinology (RV), has received more attention in recent years and has been used for the identification of vaccine proteins against different pathogens [1]. The RV approach was first applied to the bacterial pathogen Meningococcus B (MenB) and led to the license Bexsero vaccine [2], where RV played a significant role in screening for an antigen with the broadest bactericidal activity and ultimately resolved the long journey of MenB vaccine development. RV has also been applied to many other bacterial pathogens, including group A Streptococcus, antibiotic-resistant Staphylococcus aureus, Streptococcus pneumonia, and Chlamydia. The efficacy of peptide or subunit based vaccines initially identified through a RV protocol has also been proven experimentally [3,4]. In this study, a RV approach was used to screen possible vaccine proteins against the Staphylococcus hominis, identifying two extracellular proteins (N-acetylglucosaminidase and Thermonuclease family protein), as a strong candidate for vaccine development. Experimental follow up by testing the immune protection efficacy of the screened epitopes in animal models will open for experimentalists and this study will definitely speed up vaccine development process against this pathogen. This text has been highlighted in the conclusion section of the revised manuscript.
References
- Ong E, Wong MU, Huffman A, He Y. COVID-19 coronavirus vaccine design using reverse vaccinology and machine learning. bioRxiv [Preprint]. 2020 Mar 21:2020.03.20.000141. doi: 10.1101/2020.03.20.000141. Update in: Front Immunol. 2020 Jul 03;11:1581. PMID: 32511333; PMCID: PMC7239068.
- Folaranmi T, Rubin L, Martin SW, Patel M, MacNeil JR; Centers for Disease Control (CDC). Use of Serogroup B Meningococcal Vaccines in Persons Aged ≥10 Years at Increased Risk for Serogroup B Meningococcal Disease: Recommendations of the Advisory Committee on Immunization Practices, 2015. MMWR Morb Mortal Wkly Rep. 2015 Jun 12;64(22):608-12. Erratum in: MMWR Morb Mortal Wkly Rep. 2015 Jul 31;64(29):806. PMID: 26068564; PMCID: PMC4584923.
- Maione D, Margarit I, Rinaudo CD, Masignani V, Mora M, Scarselli M, Tettelin H, Brettoni C, Iacobini ET, Rosini R, D'Agostino N, Miorin L, Buccato S, Mariani M, Galli G, Nogarotto R, Nardi-Dei V, Vegni F, Fraser C, Mancuso G, Teti G, Madoff LC, Paoletti LC, Rappuoli R, Kasper DL, Telford JL, Grandi G. Identification of a universal Group B streptococcus vaccine by multiple genome screen. Science. 2005 Jul 1;309(5731):148-50. doi: 10.1126/science.1109869. Erratum in: Science. 2013 Jan 11;339(6116):141. Nardi Dei, Vincenzo [corrected to Nardi-Dei, Vincenzo]. PMID: 15994562; PMCID: PMC1351092.
- Sette A, Rappuoli R. Reverse vaccinology: developing vaccines in the era of genomics. Immunity. 2010 Oct 29;33(4):530-41. doi: 10.1016/j.immuni.2010.09.017. PMID: 21029963; PMCID: PMC3320742.
- Some parts of the manuscript are not clearly described or too simple and should be supplemented to help readers better understand the article.
Response: Thank you for valuable comment. Each section of the manuscript is revised as per reviewer instructions.
- The title of Table 6 is written incorrectly with only one bracket.
Response: Thank you for valuable comment. Corrected in the revised manuscript as per reviewer instructions.
- The schematic diagram of Fig 5 is unclear, and there is no blue mark on the diagram.
Response: Thank you for suggestion. Figure 5 is revised for quality improvement in revised manuscript.
The word of Table 4 “Transmembrane” and Table 13 “Solution” should be in the same line
Response: Thank you for valuable comment. Corrected in the revised manuscript as per reviewer instructions.
Reviewer 2 Report
This work presents an in-silico design of a broad-spectrum multi-epitope vaccine against S. hominis.
The vaccine construct is composed of antigenic, non-allergic, non-toxic multi epitopes extracted from highly virulent proteins that are part of pa?hogen core exoproteome
Many immunoinformatics, biophysical, and subtractive proteomic techniques were used in the process .
In selecting the vaccine candidate for vaccine development, tough selection criteria are used, such as these proteins should be from the core proteome, present on cell surface or are excretory in nature, non-homologous to the host and probiotic
The epitopes that are prioritized are non-allergenic, non-toxic, antigenic, water-soluble, and had higher binding affinity to B and T cell alleles.
The results of MD simulation and binding energies calculations elaborates the stable molecular configuration and minimize system energies.
Although the pan genome based reverse vaccinology is an effective method to develop multi epitope vaccine but the potency of vaccine should be studied and confirmed by in vivo and in vitro immunological methods.

Author Response
Response to Reviewer Comments
We thank the Referee for spending time and interest in our work and for helpful comments that will greatly improve the manuscript. We have checked all the general and specific comments provided by the Referee and have made all the necessary changes according to his indications. Please refer to yellow highlighted sections in the revised manuscript.
Reviewer # 3
Comments and Suggestions for Authors
This work presents an in-silico design of a broad-spectrum multi-epitope vaccine against S. hominis.
The vaccine construct is composed of antigenic, non-allergic, non-toxic multi epitopes extracted from highly virulent proteins that are part of pa?hogen core exoproteome
Many immunoinformatics, biophysical, and subtractive proteomic techniques were used in the process .
In selecting the vaccine candidate for vaccine development, tough selection criteria are used, such as these proteins should be from the core proteome, present on cell surface or are excretory in nature, non-homologous to the host and probiotic
The epitopes that are prioritized are non-allergenic, non-toxic, antigenic, water-soluble, and had higher binding affinity to B and T cell alleles.
The results of MD simulation and binding energies calculations elaborates the stable molecular configuration and minimize system energies.
Although the pan genome based reverse vaccinology is an effective method to develop multi epitope vaccine but the potency of vaccine should be studied and confirmed by in vivo and in vitro immunological methods.
Response: We understand the reviewer concern. As the scope of the paper is computational, the main objective of the work is to provide theoretical vaccine model for experimentalists to check the designed vaccine immune protective efficacy against the said pathogen in vivo. Reverse Vaccinology (RV), has received more attention in recent years and has been used for the identification of vaccine proteins against different pathogens [1]. The RV approach was first applied to the bacterial pathogen Meningococcus B (MenB) and led to the license Bexsero vaccine [2], where RV played a significant role in screening for an antigen with the broadest bactericidal activity and ultimately resolved the long journey of MenB vaccine development. RV has also been applied to many other bacterial pathogens, including group A Streptococcus, antibiotic-resistant Staphylococcus aureus, Streptococcus pneumonia, and Chlamydia. The efficacy of peptide or subunit based vaccines initially identified through a RV protocol has also been proven experimentally [3,4]. In this study, a RV approach was used to screen possible vaccine proteins against the Staphylococcus hominis, identifying two extracellular proteins (N-acetylglucosaminidase and Thermonuclease family protein), as a strong candidate for vaccine development. Experimental follow up by testing the immune protection efficacy of the screened epitopes in animal models will open for experimentalists and this study will definitely speed up vaccine development process against this pathogen. This text has been highlighted in the conclusion section of the revised manuscript.
References
- Ong E, Wong MU, Huffman A, He Y. COVID-19 coronavirus vaccine design using reverse vaccinology and machine learning. bioRxiv [Preprint]. 2020 Mar 21:2020.03.20.000141. doi: 10.1101/2020.03.20.000141. Update in: Front Immunol. 2020 Jul 03;11:1581. PMID: 32511333; PMCID: PMC7239068.
- Folaranmi T, Rubin L, Martin SW, Patel M, MacNeil JR; Centers for Disease Control (CDC). Use of Serogroup B Meningococcal Vaccines in Persons Aged ≥10 Years at Increased Risk for Serogroup B Meningococcal Disease: Recommendations of the Advisory Committee on Immunization Practices, 2015. MMWR Morb Mortal Wkly Rep. 2015 Jun 12;64(22):608-12. Erratum in: MMWR Morb Mortal Wkly Rep. 2015 Jul 31;64(29):806. PMID: 26068564; PMCID: PMC4584923.
- Maione D, Margarit I, Rinaudo CD, Masignani V, Mora M, Scarselli M, Tettelin H, Brettoni C, Iacobini ET, Rosini R, D'Agostino N, Miorin L, Buccato S, Mariani M, Galli G, Nogarotto R, Nardi-Dei V, Vegni F, Fraser C, Mancuso G, Teti G, Madoff LC, Paoletti LC, Rappuoli R, Kasper DL, Telford JL, Grandi G. Identification of a universal Group B streptococcus vaccine by multiple genome screen. Science. 2005 Jul 1;309(5731):148-50. doi: 10.1126/science.1109869. Erratum in: Science. 2013 Jan 11;339(6116):141. Nardi Dei, Vincenzo [corrected to Nardi-Dei, Vincenzo]. PMID: 15994562; PMCID: PMC1351092.
- Sette A, Rappuoli R. Reverse vaccinology: developing vaccines in the era of genomics. Immunity. 2010 Oct 29;33(4):530-41. doi: 10.1016/j.immuni.2010.09.017. PMID: 21029963; PMCID: PMC3320742.